# Deformation Mechanisms and Processing Maps for High Entropy Alloys (Presentation of Processing Maps in Terms of Zener–Hollomon Parameter): Review

**DOI:** 10.3390/ma16030919

**Published:** 2023-01-18

**Authors:** Hee-Tae Jeong, Woo Jin Kim

**Affiliations:** Department of Materials Science and Engineering, Hongik University, Mapo-gu, Sangsu-dong 72-1, Seoul 121-791, Republic of Korea

**Keywords:** high-entropy alloys, hot compression, deformation mechanisms, processing maps

## Abstract

In this review paper, the hot compressive deformation mechanisms and processing maps of high-entropy alloys (HEAs) with different chemical compositions and crystal structures are analyzed. The stress exponent (*n*_1_) values measured from the series of compression tests for the HEAs performed at different temperatures and strain rates are distributed between 3 and 35, and they are most populated between 3 and 7. Power law breakdown (PLB) is found to typically occur at *T*/*T_m_* ≤ 0.6 (where *T* is the testing temperature and *T_m_* is the melting temperature). In Al*_x_*CrMnFeCoNi (*x* = 0–1) and Al*_x_*CrFeCoNi (*x* = 0–1) HEAs, *n*_1_ tends to decrease as the concentration of Al increases, suggesting that Al acts as a solute atom that exerts a drag force on dislocation slip motion at high temperatures. The values of activation energy for plastic flow (*Q*_c_) for the HEAs are most populated in the range between 300 and 400 kJ/mol. These values are close to the activation energy of the tracer diffusivity of elements in the HEAs ranging between 240 and 408 kJ/mol. The power dissipation efficiency $\left(\eta \right)$ of the HEAs is shown to follow a single equation, which is uniquely related to *n*_1_. Flow instability for the HEAs is shown to occur near *n*_1_ = 7, implying that the onset of flow instability occurs at the transition from power law creep to PLB. Processing maps for the HEAs are demonstrated to be represented by plotting η as a function of the Zener–Hollomon parameter (*Z* = $\exp\left(\frac{Q_{c}}{RT}\right)$, where R is the gas constant). Flow stability prevails at *Z* ≤ 10^12^ s^−1^, while flow instability does at *Z* ≥ 3 × 10^14^ s^−1^.

## 1. Introduction

High-entropy alloys (HEAs) are a new family of solid-solution alloys made of four or more principal alloying elements alloyed in equiatomic or near-equiatomic concentrations (with each constituent element having a concentration from 5 to 35 atomic percent (at.%)) [1,2]. The first approach for designing an HEA was to obtain a single-phase solid solution by maximizing the mixing configurational entropy, but later, new design approaches involving multiple phases and/or intermetallics were explored [1,2,3].

Thermomechanical working at high temperatures is necessary not only to form and shape materials into components but also to produce the desired microstructures and properties of products [4,5]. For this reason, the high-temperature deformation mechanisms and hot workability of HEAs have been studied [6,7,8,9,10,11,12,13,14,15,16,17,18,19,20,21,22,23,24,25,26,27,28,29,30,31,32,33,34,35,36,37,38,39,40,41,42,43,44,45,46,47,48,49,50,51,52,53,54,55,56,57,58,59,60,61,62,63,64,65,66,67,68,69,70,71,72,73,74,75,76,77,78,79,80,81,82,83,84,85,86,87,88,89,90,91,92,93,94,95,96,97,98,99,100]. Finding the constitutive equations that mathematically depict the material response to the applied hot deformation conditions can be useful in predicting the flow stress or strain rates and identifying the rate-determining deformation mechanisms at different temperatures/strain rates and microstructural parameters. Characterization of hot workability using a processing map is important for the optimization of hot working conditions of materials and fabrication of defect-free components [101]. If a material is processed under unstable flow conditions, adiabatic shear banding or cracking can occur, and if a material is processed under optimal conditions, superior microstructure and mechanical properties can be obtained.

In this work, we have reviewed and analyzed the hot compression data of HEAs available in the literature to elucidate their deformation mechanisms and optimal hot working conditions.

## 2. History and Materials

The first paper reporting the hot compression behavior of HEAs was available in 2011, and the first studied HEAs were Nb_25_Mo_25_Ta_25_W_25_ and V_20_Nb_20_Mo_20_Ta_20_W_20_ alloys [6], where the microstructural change during hot compression at temperatures of 1073–1873 K and at a strain rate of 10^−3^ s^−1^ was examined. The papers published in the literature from 2011 to the present can be categorized into three groups (Figure 1a). The first group of papers reports the hot compressive deformation data of the HEAs in the limited temperature and strain rate range. The second group of papers reports the hot compression data of the HEAs in a wide range of temperature and strain rates, but processing maps are not constructed. The third group of papers reports the hot compression data as well as the processing maps of the HEAs. The number of publications increases almost exponentially with time, indicating that attention to this academic and engineering field has rapidly increased.

The HEA materials studied by hot compression can be classified into three groups (Figure 1b). The first material group is the Cr-Mn-Fe-Co-Ni series HEAs containing Al, Sn, Zr, Sn, C, and N [9,20,23,27,34,37,38,42,45,60,62,63,64]; the second material group is the Cr-Fe-Co-Ni series HEAs containing Zr, Ta, Nb, Mo, Cu, C, and N [13,25,51,53,72,84,92,96,97,100]; and the third material group is the other composition HEAs, including the materials of TiVNbMoTa, Mn_5_Co_25_Fe_25_Ni_25_Ti_20_, MnFeCoNiCu, etc. [8,11,12,22,29,32,33,40,41,47,52,56,68,71,74,79,91,93,95,99]. Information regarding the chemical compositions of the HEAs, grain sizes, crystal structure, types of phases, temperature and strain rate ranges for hot compression tests are provided in Table 1. Most of the HEAs studied for hot compression are as-cast or heat-treated (homogenized) cast with coarse grain sizes. Among the Cr-Mn-Fe-Co-Ni series HEAs and Cr-Fe-Co-Ni series HEAs, Al*_x_*CoCrFeNi and Al*_x_*CoCrFeMnNi (*x* = 0–1) HEAs [9,13,23,25,27,34,42,45,51,53,62,63,71,84,92,96,97,100] have been the most studied, where the addition of Al can facilitate the formation of BCC phase from the FCC matrix. At low Al levels corresponding to *x* = 0–0.4 or 0.5, the alloys have a single FCC phase, but with a further increase in Al content in the range of *x* = 0.4 or 0.5–0.9, both FCC and BCC phases coexist, and at Al addition beyond *x* = 0.9–0.95, a BCC single phase is obtained [102,103].

HEAs with a single FCC phase or FCC (major) + BCC (minor) phases [9,13,18,22,23,25,27,29,32,34,37,41,45,47,52,53,56,60,61,62,63,72,84,91,92,96,97,100] have been most extensively studied for hot compression tests (Figure 1c). HEAs with a single BCC phase or BCC (major) + FCC (minor) phases have also been popularly studied [8,11,33,40,42,51,68,71,74,79,93,99]. HEAs with FCC1 and FCC2 or FCC+ intermetallic compound (IMC) precipitates have been recently studied [12,20,38,54,64,66,70,89,90,95].

## 3. Deformation Mechanisms

The raw data (true stress–true strain curves) in the second and third groups of papers, where a series of hot compression tests were systematically carried out over wide temperature and strain rate ranges, were digitally extracted from the published papers using a software. For example, the true stress–true strain curves for Al_0.7_CrMnFeCoNi HEA [62] at different temperatures and different strain rates are shown in Figure 2a–d.

The hyperbolic sine Garofalo equation has been widely used to describe the steady-state relationship between the flow stress, temperature, and strain rate ($\dot{\varepsilon }$) over a wide range of temperatures and strain rates where power law creep and power law breakdown (PLB) govern plastic flow [4,104], which is expressed as:(1)$$\dot{\varepsilon }=A\;\sin h{\left(\alpha \;\sigma \right)}^{n}\exp\left(-\frac{Q_{c}}{RT}\right)$$
where *A* and *α* are the experimentally determined material constants, *n* is the stress exponent, and *Q*_c_ is the activation energy for plastic flow. The hyperbolic sine function is mathematically reduced to the equation describing power law creep at low stresses (Equation (2)) and to the equation describing power law breakdown (PLB) at high stresses (Equation (3)):(2)$$\dot{\varepsilon }=A_{1}\;{\left(\sigma \right)}^{n_{1}}\exp\left(-\frac{Q_{c}}{RT}\right)$$
(3)$$\dot{\varepsilon }=A_{2}\;\exp\left(\beta \;\sigma \right)\;\exp\left(-\frac{Q_{c}}{RT}\right)$$
where $A_{1}$, $A_{2}$, and *β* are the material constants and *n*_1_ is the stress exponent, which is ideally equal to *n,* but can be different). The constants *α* in Equation (1), *n*_1_ in Equation (2), and *β* in Equation (3) can be related by *β* = *αn*_1_ [4,104].

Figure 3a,b shows the plots of $\log\dot{\varepsilon }-\log\sigma$ and $\log\dot{\varepsilon }-\sigma$ at a given strain of 0.7 for the Al_0.7_CrMnFeCoNi HEA, where steady-state (SS) flow is attained at almost all temperatures and strain rates. The slopes of the regression lines of the $\log\dot{\varepsilon }-\log\sigma$ and $\log\dot{\varepsilon }-\sigma$ curves are used to determine the values of *n*_1_ and *β* at each temperature, respectively. Figure 3c shows the plot of $\log\dot{\varepsilon }-\log\left(\sinh\left(\alpha \;\sigma \right)\right)$, where α is the average of the $\alpha \left(=\beta /n_{1}\right)$ values measured at all temperatures. According to Equation (1), the slopes (=${\left[\frac{\partial \log\dot{\varepsilon }}{\partial \log\left(\sinh\left(\alpha \;\sigma \right)\right)}\right]}_{T})$ of the $\log\dot{\varepsilon }-\log\left(\sinh\left(\alpha \;\sigma \right)\right)$ curves represent the *n* value at each temperature. Figure 3d shows the plot of $\log\left(\sinh\left(\alpha \;\sigma \right)\right)-\frac{1000}{T}$, of which the slope (=${\left[\frac{\partial \ln\left(\sinh\left(\alpha \;\sigma \right)\right)}{\partial \left(\frac{1000}{T}\right)}\right]}_{\dot{\varepsilon }})$ represents the *s* value at each strain rate. The *Q*_c_ value can be calculated by using the average (*N*) of the *n* values measured at different temperatures and the average (*S*) of the *s* values measured at different strain rates as follows:(4)$$Q_{c}=RNS$$

It is well-known that a value of *n*_1_ represents a specific deformation mechanism; Haper–Dorn creep and diffusional creep are associated with *n*_1_ = 1 [105,106,107,108], grain boundary sliding is associated with *n*_1_ = 2 [109,110], solute drag creep is associated with *n*_1_ = 3 [111], dislocation climb creep is associated with 5–7 [112,113], and power law breakdown occurs when *n*_1_ > 7 [4]. Jeong and Kim [62] analyzed the tensile, compressive, and creep data of CrMnFeCoNi and Al_0.5_CrMnFeCoNi HEAs with various grain sizes and proposed the constitutive equations that can quantitatively predict their flow stresses as a function of strain rate, temperature, and grain size (Table 2). Figure 4a,b shows the *n*_1_ values of HEAs with different crystal structures as a function of temperature (*T*) and a homologous temperature (*T*/*T_m_*), respectively, where *T_m_* is the melting temperature of the HEAs. As the *T_m_* values of most of the HEAs are not available in the literature, we calculated the *T_m_* values of the HEAs using the relation $T_{m}=\overset{n}{\underset{i=1}{\sum }}c_{i}{\left(T_{m}\right)}_{i}$ [114], where $c_{i}$ is the atomic percentage of the *i*th component and ${\left(T_{m}\right)}_{i}$ is the melting point of the *i*th component of the alloy. The plots in Figure 4a,b show that the hot compression tests for the HEAs have been conducted in the temperature range between 873 K and 1573 K, corresponding to the *T*/*T_m_* range between 0.4 and 0.9. The *n*_1_ value is distributed between 3 and 35, and the *n*_1_ values are most populated between 3 and 7. As *T* and *T*/*T_m_* decrease, *n*_1_ tends to increase, and at *T*/*T_m_* ≤ 0.6, the number of the data associated with *n*_1_ ≥ 7 sharply increases. In the *T*/*T_m_* range between 0.6 and 0.9, the fraction of the data associated with *n*_1_ ≥ 7 is 0.17, while in the *T*/*T_m_* range between 0.4 and 0.6, the fraction of *n*_1_ ≥ 7 is 0.66. This result indicates that PLB dominates plastic flow below 0.6 *T*/*T_m_*, while power law creep dominates plastic flow above 0.6 *T*/*T_m_*. It should be noted that the HEAs that show the lowest *n*_1_ of ~3 are the Al_x_CoCrFeNi and Al_x_CoCrFeMnNi HEAs with BCC or BCC + FCC (minor) phases, containing an Al element. Figure 4c shows that the *n* values of the HEAs are slightly smaller than their *n*_1_ values. They mostly range between 2.5 and 5. The *n* tends to increase as *T*/*T_m_* decreases, but it is not as sensitive as *n*_1_ to *T*/*T_m_* variation.

Figure 5a,b shows the $\log\dot{\varepsilon }-\log\sigma$ curves for the Al*_x_*CrMnFeCoNi (*x* = 0–1) [9,27,34,42,63] and Al*_x_*CrFeCoNi (*x* = 0–1) HEAs [13,25,51,53,72,84,92,96] at a given temperature of 1173 K. The *n*_1_ value decreases from 5 to ~3 as *x* increases. This result indicates that as the amount of BCC phase (rich with Al) increases, the characteristics of viscous glide creep associated with *n*_1_~3 become more pronounced. Jeong and Kim [42] analyzed the deformation behavior of the AlCrMnFeCoNi HEA and found that aluminum, which is largest in size among the constituent elements [42], acts as a solute that causes solute drag creep and showed that the solute drag creep model proposed by Hong and Weertman [111] for conventional metals can quantitatively explain the deformation behavior of the AlCrMnFeCoNi HEA with reasonable assumptions on the diffusivity of Al in the HEA. From the plots in Figure 5a,b, it should also be noted that flow stress tends to decrease the added amount of Al increases especially at low strain rates, indicating that the BCC phase deforming under solute drag creep is weaker than the FCC phase deforming under dislocation climb creep.

Deformation mechanism maps represent the dominant deformation mechanism for a given metallic material under different conditions. Figure 6a,b shows the deformation mechanism maps as a function of strain rate (10^−5^ to 10 s^−1^) and temperature (873 and 1573 K) at a given (coarse) grain size of 100 μm for CoCrFeMnNi and Al_0.5_CoCrFeMnNi. On the maps, the data of the CoCrFeMnNi and Al_0.5_CoCrFeMnNi HEAs are loaded. For the CoCrFeMnNi HEA, at high temperatures, dislocation climb creep controlled by lattice diffusivity (*D*_L_) governs plastic flow, but when the temperature is low, the rate-controlling mechanism changes to dislocation climb creep controlled by *D*_p_ (or PLB). As the strain rate increases, the region associated with *D*_p_-controlled dislocation climb creep (or PLB) expands to a higher temperature. For the Al_0.5_CoCrFeMnNi, solute drag creep appears in the bottom right corner of the map. Jeong and Kim [62] showed that when the grain size is sufficiently small, grain boundary sliding mechanism can play a more important role than solute drag creep in the Al_0.5_CoCrFeMnNi HEA.

Figure 7a shows the *Q*_c_ values of HEAs calculated using Equation (4). The *Q*_c_ value is in the range between 150 and 600 kJ/mol, and the data distribution is most populated in the range between 300 and 400 kJ/mol. The activation energy of the tracer diffusivity of elements in the HEAs ranges between 240 and 408 kJ/mol [115,116], implying that the activation energy of plastic flow for the HEA is related to the atomic diffusivity of elements constituting the HEAs. Figure 7b shows the relation between *n*_1_ and *Q*_c_ for the HEAs. A smaller *Q*_c_ is obtained at smaller *n*_1_, and this is more apparent near *n*_1_~3, where solute drag creep governs the deformation mechanism. It is worthwhile to note that the activation energy for the solute diffusion (*Q_solute_*) of magnesium in aluminum (136 kJ mol^−1^) is lower than the activation energy for self-diffusion in pure aluminum (142 kJ/mol) [117].

## 4. Processing Maps

A processing map, which is useful in finding the optimal condition for hot forging or extrusion, is composed of a power dissipation map and a flow instability map. According to Prasad et al. [101], the total power, *P*, absorbed in a material is divided into the power dissipation content (*G*), which represents the power dissipated by plastic deformation giving rise to a temperature increase in the workpiece and the power dissipation co-content (*J*), which represents the power dissipated by a change in its microstructure, such as dynamic recovery and dynamic recrystallization [101].
(5)$$P=\sigma \dot{\varepsilon }=G+J=\int_{0}^{\dot{\varepsilon }}\sigma d\dot{\varepsilon }+\int_{0}^{\sigma }\dot{\varepsilon }d\sigma$$

The efficiency of power dissipation, *η*, which represents the power dissipation efficiency due to a change in microstructure during plastic flow, is defined as [101]:(6)$$\eta =\frac{J}{J_{max}}=2\left(1-\frac{1}{\sigma \dot{\varepsilon }}\int_{0}^{\dot{\varepsilon }}\sigma d\dot{\varepsilon }\right)$$
where *J_max_* is the maximum *J* value (=*P*/2).

The strain rate sensitivity, *m*, which is equal to 1/*n*_1_, can be calculated by:(7)$$m=\frac{1}{n_{1}}=\left[\frac{\partial ln\sigma }{\partial ln\dot{\varepsilon }}\right]$$

When *m* is assumed to be constant over the investigated strain rate range (as assumed by Prasad et al. [101]), $\eta =\frac{2m}{m+1}$, but when *m* is not constant (as considered by Murty et al. [118]), *η* can be directly determined from Equation (6) by calculating $\int_{0}^{\dot{\varepsilon }}\sigma d\dot{\varepsilon }$ through the numerical integration procedure, using Equation (8):(8)$$\int_{0}^{\dot{\varepsilon }}\sigma d\dot{\varepsilon }=\int_{0}^{{\dot{\varepsilon }}_{min}}\sigma d\dot{\varepsilon }+\int_{{\dot{\varepsilon }}_{min}}^{\dot{\varepsilon }}\sigma d\dot{\varepsilon }={\left(\frac{\sigma \dot{\varepsilon }}{m+1}\right)}_{{\dot{\varepsilon }}_{min}}+\;\int_{{\dot{\varepsilon }}_{min}}^{\dot{\varepsilon }}\sigma d\dot{\varepsilon }$$

In drawing the flow instability map, Ziegler’s plastic flow theory is used, and according to Murty et al. [119] (when *m* is not a constant),
(9)ξ=2m−η<0

When *ξ* is negative, deformation in the material is predicted to be unstable, such that localized flow, adiabatic shear banding, or cracking can take place.

Kim and Jeong [120] suggested an empirical equation for η by analyzing the behavior of η values of many metals calculated by following the numerical method proposed by Murty (Equation (8)) as a function of $n_{1}$:(10)$$\eta =\left[\frac{10^{4}}{{\left(n_{1}-1\right)}^{2}}\right]\left(\frac{2}{n_{1}+1}\right)+\left[\frac{\tanh\left(n_{1}-5\right)+1}{2}\cdot \frac{{\left(n_{1}-1\right)}^{1.5}}{{\left(n_{1}-1\right)}^{1.5}+10^{2}}\right]\left(\frac{2}{n_{1}}\cdot \frac{e^{n_{1}}-1}{e^{n_{1}}}\right)$$

By using this equation, the η value can be easily obtained once *n*_1_ is known without numerically solving Equation (8). Kim and Jeong [121] also presented a simple form of the flow instability criterion based on the observation that unstable flow occurs in many metals when $n_{1}$ is larger than 7 (i.e. when PLB governs plastic flow):(11)$$\xi =7-n_{1}<0\;\;\mathrm{or}\;\eta <0.285$$

Unlike in the procedure for determining $n_{1}$ for calculating $Q_{c}$ (Figure 3a), where a linear fitting is applied to the data in the plot of $\log\dot{\varepsilon }-\log\sigma$, the $n_{1}$ value as a function of strain rate, temperature, and strain, which is necessary for constructing processing maps, has often been determined using a third- or fourth-order polynomial fitting to the data in the plot of $\log\dot{\varepsilon }-\log\sigma$. This polynomial fitting curve, however, sometimes has difficulty describing the power law creep. This example is shown in Figure 8. Thus, Kim and Jeong [121] proposed the exponential fitting method for the determination of the *m* from the plot of $\log\dot{\varepsilon }-\log\sigma$. According to the method,
(12)$$\log\dot{\varepsilon }=a+c\;\times \;\exp\left(\frac{\log\sigma -b}{d}\right)+f\times \exp\left(\frac{\log\sigma -b}{g}\right)$$

When Equation (12) is used, *m* can be calculated by Equation (13):(13)$$m={\left[\frac{c}{d}\times \exp\left(\frac{\log\sigma -b}{d}\right)+\frac{f}{g}\times \exp\left(\frac{\log\sigma -b}{g}\right)\right]}^{-1}$$

As observed in Figure 8, the exponential fitting, where *m* tends to decrease gradually with increasing strain rate, provides a better fit to the series of data compared with the polynomial fitting, which sometimes creates uncertain fluctuation between the data points. 

Figure 9a–c shows the *η* values of the HEAs calculated by Murty’s method (Equations (6) and (8)) as a function of *n*_1_ for the three material groups of HEAs. It is obvious that the *η* values of all the three material groups of HEAs follow Equation (10) well in the entire range of *n*_1_, regardless of the differences in composition and crystal structure. Furthermore, it is observed that the flow instability condition determined by Equation (9) occurs at *n*_1_≈7, supporting that the onset of flow instability occurs at the transition from power law creep to PLB (Equation (11)).

Figure 10a–d shows the processing maps for Al_0.5_CrMnFeCoNi and Al_0.3_CrFeCoNi HEAs constructed based on Murty’s approach and Kim and Jeong’s approach. A good match is observed between the two methods in power dissipation maps as well as flow instability maps. However, some mismatch is observed for the Al_0.3_CrFeCoNi HEA in the flow instability at low strain rates and at low temperatures. This occurs because in Murty’s method, the material is assumed to follow the power law creep at low strain rates below ${\dot{\varepsilon }}_{min}$ (Equation (8)), but this assumption can be wrong at low temperatures if PLB governs plastic flow below ${\dot{\varepsilon }}_{min}$.

Figure 11a,b shows the plots of *n*_1_ as a function of the Zener–Hollomon parameter (*Z* = $\exp\left(\frac{Q_{c}}{RT}\right))$ for Al_0.5_CrMnFeCoNi and Al_0.3_CrFeCoNi. There is a good correlation between *n*_1_ and Z, indicating that as strain rate increases and temperature decreases (i.e., as *Z* increases), *n*_1_ tends to increase. This occurs because according to the deformation mechanism maps (Figure 6a,b), as strain rate increases and temperature decreases, the deformation mechanism changes from solute drag creep (associated with *n*_1_ = 3) to dislocation climb creep (associated with *n*_1_ = 5) and then power law breakdown (associated with *n*_1_ > 3). η is a function of *n*_1_ according to Equation (10). Hence, η can also be expressed as a function of Z. Figure 11c,d shows the plot of η as a function of *Z* for Al_0.5_CrMnFeCoNi and Al_0.3_CrFeCoNi, where a good correlation between η and *Z* is observed. η tends to decrease as *Z* increases. In addition, most of the data belonging to the flow instability condition (determined by Murty’s approach using Equation (9)) are positioned below η=0.285, supporting the validity of Equation (11). Some mismatches are observed between Equation (9) and Equation (10) and this can be attributed to the aforementioned assumption of power law creep below ${\dot{\varepsilon }}_{min}$ in Murty’s method.

The plots in Figure 11c,d represent the “processing maps expressed as a function of *Z*” because if one knows the temperature and strain rate, the *Z* value can be calculated, and then the power dissipation efficiency and flow (in)stability can be readily determined from the plot.

Figure 12a–c shows the plots of processing maps for the three material groups of HEAs expressed as a function of *Z* using each *Q*_c_ value of the HEAs (Table 1), and Figure 12d shows the plot where all the data in Figure 12a–c overlap. The *η* values of each material are well correlated as a function of Z, and the data for each group converge to a common curve. Also, it is observed that all the data of the three groups converge to a single common curve. According to the plot in Figure 12d, flow stability is nearly guaranteed at *Z* ≤ 10^12^ s^−1^, while flow instability is nearly inevitable at *Z* ≥ 2 × 10^15^ s^−1^. At 10^12^ s^−1^ ≤ *Z* ≤ 2 × 10^15^ s^−1^, flow stability and instability conditions coexist, and flow instability becomes more dominant as *Z* increases. 

Figure 13a–c shows the plots of processing maps for the three material groups of HEAs constructed as a function of *Z* using the average *Q*_c_ value of all the HEAs (317.2 kJ/mol), and Figure 12d shows the plot where all the data in Figure 13a–c overlap. Note that all the data lie close to a common curve. According to the plot in Figure 12d, flow stability prevails at *Z* ≤ 10^12^ s^−1^, while flow instability prevails at *Z* ≥ 3 × 10^14^ s^−1^. By plotting in this fashion, one can easily compare the *η* values of the different HEAs at a given temperature and strain rate as well as predict the optimum hot working conditions of the HEAs with unknown *Q*_c_ values. 

## 5. Conclusions

The hot compressive behaviors of the HEA materials with different chemical compositions and crystal structures and processing maps were analyzed, and the following observations were made.

Hot compression tests on many HEAs have been conducted in the temperature range between 873 K and 1573 K, corresponding to the *T*/*T_m_* range between 0.4 and 0.9. The *n*_1_ values are most populated between 3 and 7.As *T*/*T_m_* decreases, *n*_1_ tends to increase, and power law breakdown typically occurs at *T*/*T_m_* ≤ 0.6.In Al*_x_*CrMnFeCoNi (*x* = 0–1) and Al*_x_*CrFeCoNi (*x* = 0–1) HEAs, *n*_1_ tends to increase as the concentration of Al increases, implying that Al acts as a solute atom that exerts a drag force on dislocation slip motion.The activation energy for plastic flow (*Q*_c_) in the HEAs is calculated to be in the range between 150 and 600 kJ/mol, and the data distribution is populated in the *Q*_c_ value range between 300 and 400 kJ/mol. The average *Q*_c_ value for all the HEAs is 317 kJ/mol.The η value of the HEAs can be expressed as a function of *n*_1_ only. Flow instability is shown to occur near *n*_1_ = 7, implying that the onset of flow instability occurs at the transition from power law creep to PLB.Processing maps for all the HEAs are demonstrated to be constructed using the Zener–Hollomon parameter (*Z* = $\exp\left(\frac{Q_{c}}{RT}\right))$. According to the analysis result, flow stability prevails at *Z* ≤ 10^12^ s^−1^ in all HEAs.

## Figures and Tables

**Figure 1 materials-16-00919-f001:**
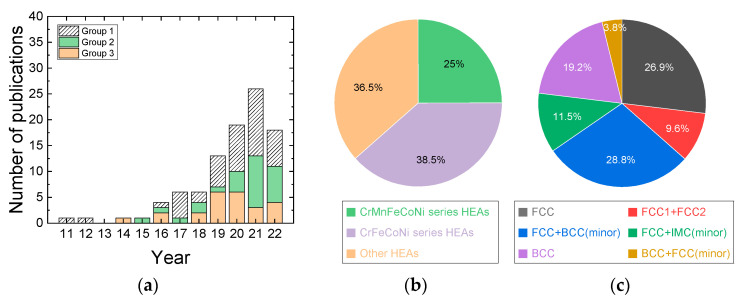
(**a**) Papers published in the literature from 2011 are categorized into three groups. The first group of papers includes papers that report hot compressive deformation of the HEAs in the limited temperature and strain rate range, the second group of papers provides hot compression data of the HEAs over a wide range of temperature and strain rates, but processing maps are not constructed, and the third group of papers provides hot compression data (over a wide range of temperature and strain rates) as well as processing maps of the HEAs. (**b**) Three material groups of HEA materials studied for hot compression, which are classified by their chemical compositions. (**c**) HEA materials studied for hot compression, which are classified by phases (crystal structures).

**Figure 2 materials-16-00919-f002:**
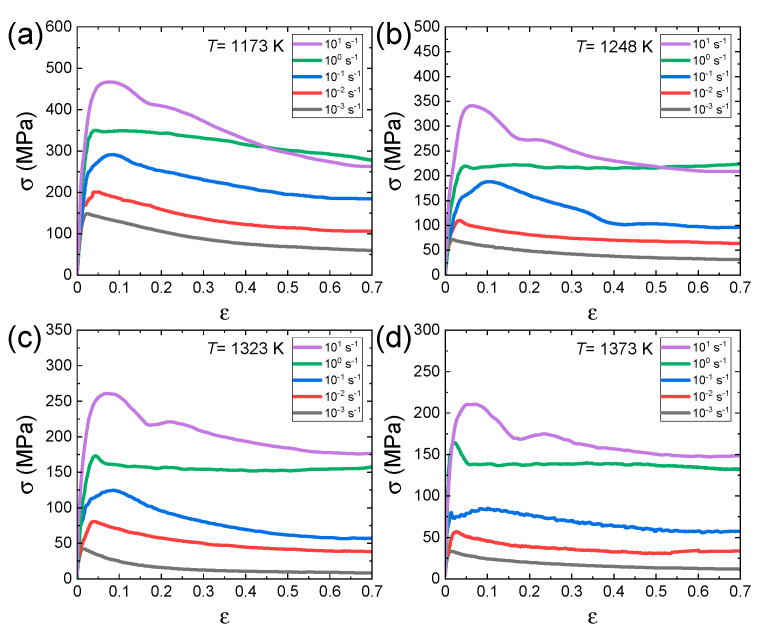
True stress–strain curves for Al_0.7_CrMnFeCoNi HEA at various strain rates at temperatures of (**a**) 1173, (**b**) 1248, (**c**) 1323, and (**d**) 1373 K. Reproduced/modified with permission from [62], Elsevier.

**Figure 3 materials-16-00919-f003:**
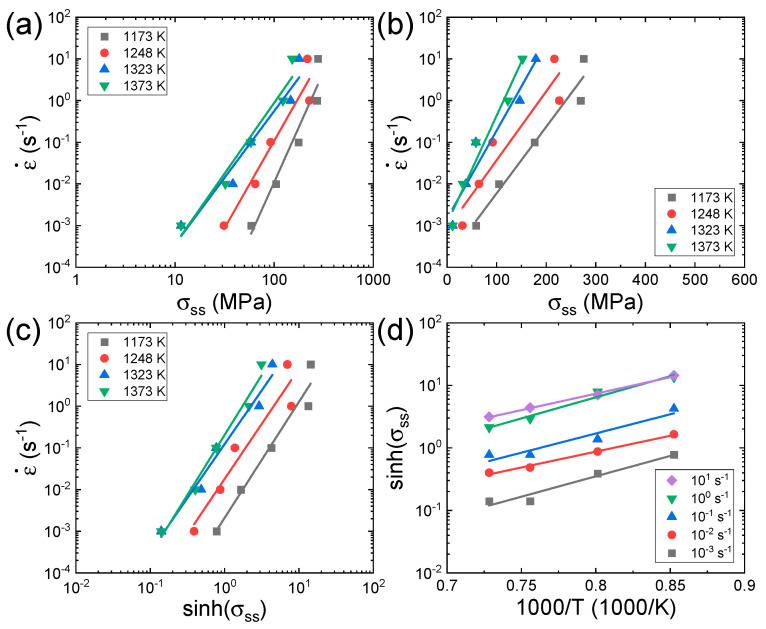
Plots of (**a**) $\log\dot{\varepsilon }-\log\sigma$, (**b**) $\log\dot{\varepsilon }-\sigma$, (**c**) $\log\dot{\varepsilon }-\log\left(\sinh\left(\alpha \;\sigma \right)\right)$, and (**d**) $\log\left(\sinh\left(\alpha \;\sigma \right)\right)-\frac{1000}{T}$ for the Al_0.7_CrMnFeCoNi HEA [62] at a given strain of 0.7. The subscript ‘*ss*’ in *σ_ss_* represents the steady state. Reproduced/modified with permission from [62], Elsevier.

**Figure 4 materials-16-00919-f004:**
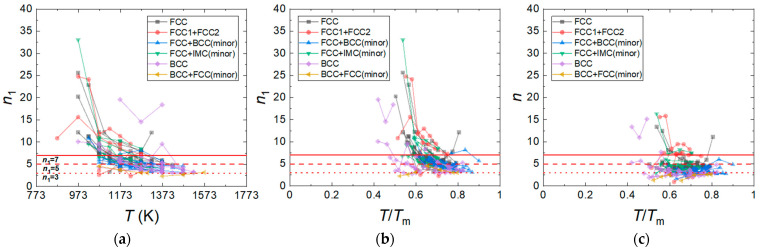
*n*_1_ values of the HEAs with different crystal structures as a function of (**a**) temperature (*T*) and (**b**) homologous temperature (*T*/*T_m_*). (**c**) *n* values of the HEAs as a function of *T*/*T_m_*.

**Figure 5 materials-16-00919-f005:**
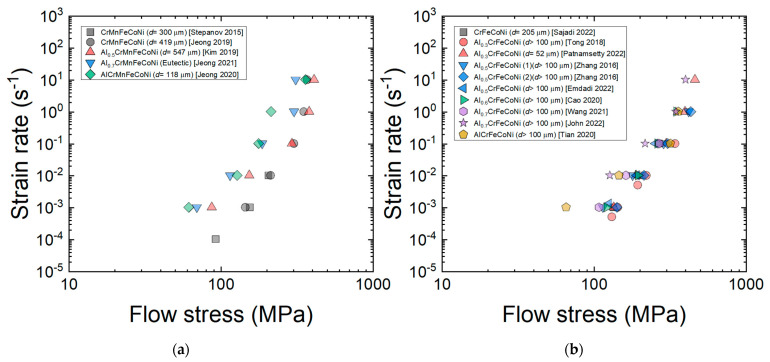
Plots of the $\log\dot{\varepsilon }-\log\sigma$ curves for the (**a**) Al*_x_*CrMnFeCoNi (*x* = 0–1) [9,27,34,42,63] and (**b**) Al*_x_*CrFeCoNi (*x* = 0–1) [13,25,51,53,72,84,92,96] HEAs at a given temperature of 1173 K (at a strain of 0.5).

**Figure 6 materials-16-00919-f006:**
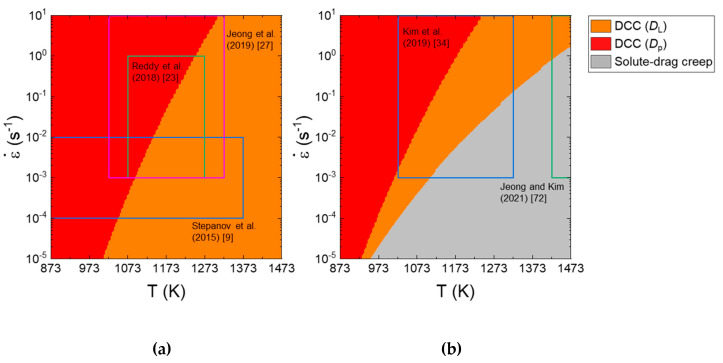
Deformation mechanism maps as a function of strain rate and temperature at a given grain size of 100 μm for (**a**) CoCrFeMnNi and (**b**) Al_0.5_CoCrFeMnNi. Reproduced/modified with permission from [62], Elsevier.

**Figure 7 materials-16-00919-f007:**
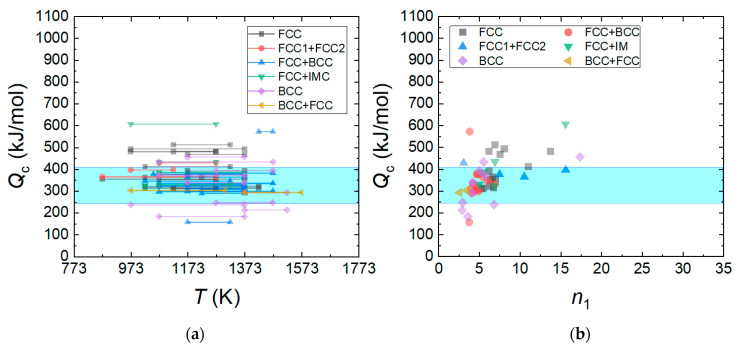
(**a**) *Q*_c_ values of HEAs calculated using Equation (4) and (**b**) the relation between *n*_1_ and *Q*_c_ for the HEAs. The activation energy ($Q_{L}^{*}$) of tracer diffusivity of elements in the Al*_x_*CoCrFeMnNi and Al*_x_*CoCrFeNi HEAs ranges between 240 and 408 kJ/mol (shaded area by blue color): $Q_{L}^{*}$ = 323 ± 5 kJ/mol for Cr, $Q_{L}^{*}$ = 303 ± 3 kJ/mol for Fe, $Q_{L}^{*}$ = 240 ± 20 kJ/mol for Co, $Q_{L}^{*}$ = 253 ± 8 kJ/mol for Ni in CrFeCoNi, $Q_{L}^{*}$ = 313 ± 13 kJ/mol for Cr, $Q_{L}^{*}$ = 272 ± 13 kJ/mol for Mn, $Q_{L}^{*}$ = 309 ± 11 kJ/mol for Fe, $Q_{L}^{*}$ = 270 ± 22 kJ/mol for Co, and $Q_{L}^{*}$ = 304 ± 9 kJ/mol for Ni in CrMnFeCoNi [115]. $Q_{L}^{*}$ = 263 kJ/mol for Al, $Q_{L}^{*}$ = 288 kJ/mol for Cr, $Q_{L}^{*}$ = 315 kJ/mol for Fe, $Q_{L}^{*}$ = 258 kJ/mol for Co, $Q_{L}^{*}$ = 260 kJ/mol for Ni in Al_4.88_Co_29.53_Cr_18.58_Fe_19.62_Ni_27.39_ [117], $Q_{L}^{*}$ = 258 kJ/mol for Al, $Q_{L}^{*}$ = 288 kJ/mol for Cr, $Q_{L}^{*}$ = 408 kJ/mol for Fe, $Q_{L}^{*}$ = 260 kJ/mol for Co, and $Q_{L}^{*}$ = 261 kJ/mol for Ni Al_6.64_Co_23.82_Cr_23.66_Fe_23.01_Ni_22.87_ [116].

**Figure 8 materials-16-00919-f008:**
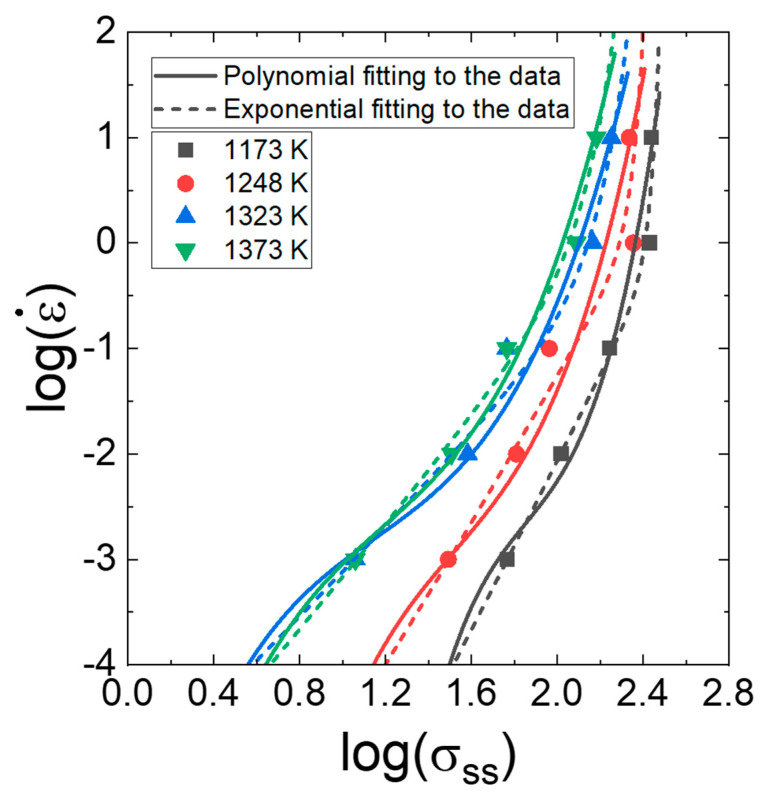
Polynomial fitting and exponential fitting to the data points in the plot in Figure 3a. Reproduced/modified with permission from [63], Elsevier.

**Figure 9 materials-16-00919-f009:**
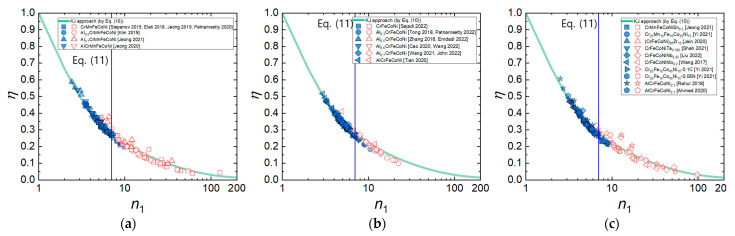
*η* values of the HEAs calculated by Murty’s method as a function of *n*_1_ for (**a**) Al*_x_*CrMnFeCoNi (*x* = 0–1) [9,23,27,34,42,45,63], (**b**) Al*_x_*CrFeCoNi (*x* = 0–1) [13,25,51,53,72,84,92,96,97,100], and (**c**) CrMnFeCoNiSn_0.5_, etc. [18,22,47,54,60,61,64,70,89], which can be fitted by Equation (10). The (blue) solid and (red) open symbols represent the data points belonging to the flow stability and flow instability regimes, respectively, which is determined by Equation (9).

**Figure 10 materials-16-00919-f010:**
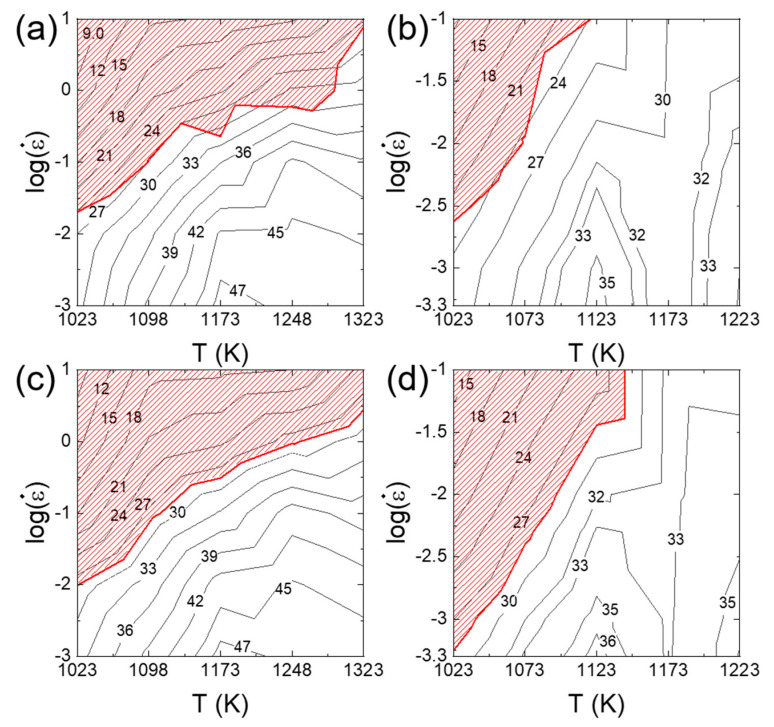
2D processing maps for (**a**,**c**) Al_0.5_CrMnFeCoNi and (**b**,**d**) Al_0.3_CrFeCoNi constructed based on Murty’s approach and Kim and Jeong’s approach using the raw data from [62] (for Al_0.5_CrMnFeCoNi) and [92] (for Al_0.3_CrFeCoNi).

**Figure 11 materials-16-00919-f011:**
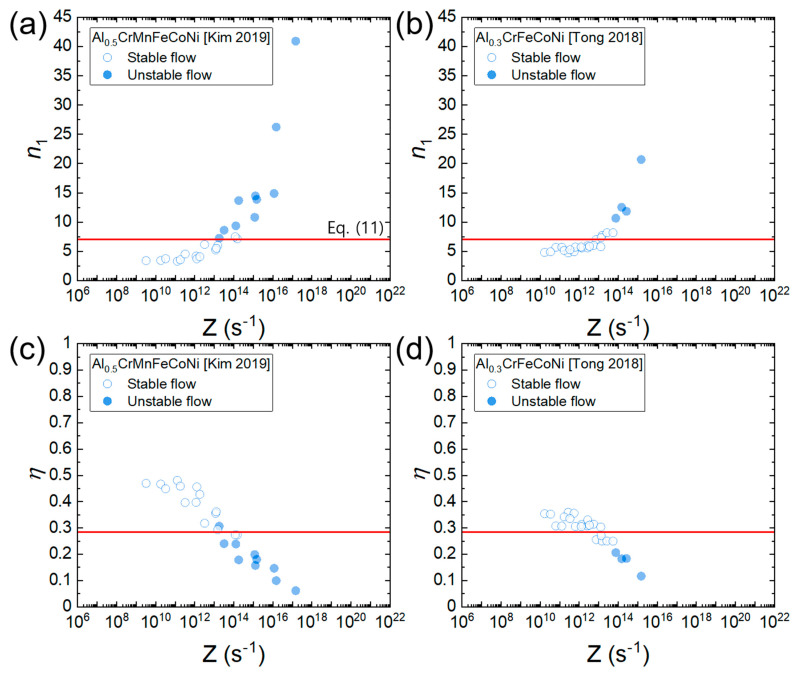
Plot of *n*_1_ and η as a function of *Z* for (**a**,**c**) Al_0.5_CrMnFeCoNi [34] and (**b**,**d**) Al_0.3_CrFeCoNi [25]. Open and solid symbols represent the flow stability and instability conditions (determined by Equation (9)), respectively. A red horizontal line represents η=0.285 (Equation (11)).

**Figure 12 materials-16-00919-f012:**
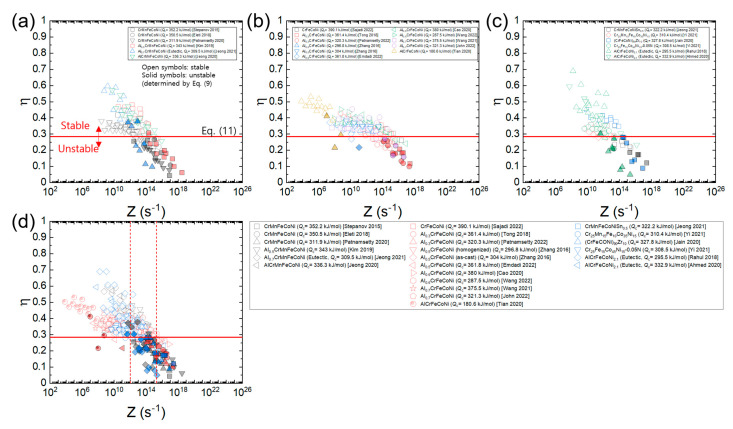
Plots of processing maps for (**a**) Al*_x_*CrMnFeCoNi (*x* = 0–1) [9,23,34,42,45,63], (**b**) Al*_x_*CrFeCoNi (*x* = 0–1) [13,25,51,53,72,84,92,96,97,100], and (**c**) CrMnFeCoNiSn_0.5_, etc. [22,47,54,60,61,64], expressed as a function of *Z* using each *Q*_c_ value of the HEAs (Table 1) and (**d**) the plot where all the data in (**a**–**c**) overlap. Open and solid symbols represent the flow stability and instability conditions (determined by Equation (9)), respectively. A red horizontal line represents η=0.285 (Equation (11)).

**Figure 13 materials-16-00919-f013:**
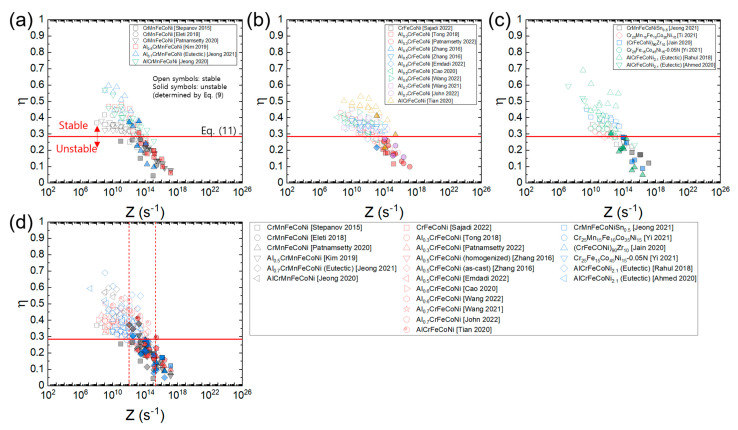
Plots of processing maps for (**a**) Al*_x_*CrMnFeCoNi (x = 0–1) [9,23,34,42,45,63], (**b**) AlxCrFeCoNi (x = 0–1) [13,25,51,53,72,84,92,96,97,100], and (**c**) CrMnFeCoNiSn0.5, etc. [22,47,54,60,61,64], expressed as a function of *Z* using the average Qc value of all the HEAs (317.2 kJ/mol) and (**d**) the plot where all the data in (**a**–**c**) overlap. Open and solid symbols represent the flow stability and instability conditions (determined by Equation (9)), respectively. A red horizontal line represents η=0.285 (Equation (11)).

**Table 1 materials-16-00919-t001:** Information regarding the chemical compositions of the HEAs, grain sizes, crystal structure, types of phases, temperature and strain rate ranges for hot compression tests.

Composition	Grain Size (μm)	Major Phase	Minor Phase	*T* Range (K)	$\dot{\varepsilon }$ Range (s^−1^)	*T*_m_ (K)	*n* _1_	*Q*_c_ (kJ/mol)	Material Group
CrMnFeCoNi [9]	300	FCC	-	873–1373	10^−4^–10^−2^	1801	4.9–12.1	352.2 (*ε*_SS_)	1
CrMnFeCoNi [23]		FCC	-	1073–1273	10^−3^–1	1801	5–8.5	350.5 (*ε*_0.9_)	1
CrMnFeCoNi [27]	419	FCC	-	1023–1323	10^−3^–10	1801	5.8–22.8	410.9 (*ε*_0.6_)	1
CrMnFeCoNi [45]	12.8	FCC	-	1023–1423	10^−3^–10	1801	4.7–11.1	311.9 (*ε*_0.7_)	1
Al_0.5_CrMnFeCoNi [34]	547	FCC	BCC	1023–1323	10^−3^–10	1722	4.2–10.8	343 (*ε*_0.6_)	1
Al_0.5_CrMnFeCoNi [62]	547	FCC	BCC	1423&1473	10^−3^–10	1722	3.6–4.1	570.5 (*ε*_0.6_)	1
Al_0.7_CrMnFeCoNi [63]		FCC	BCC	1173–1373	10^−3^–10	1695	3.2–5.3	309.5 (*ε*_0.7_)	1
AlCrMnFeCoNi [42]	118	BCC	-	1173–1373	10^−3^–10	1657	3.4–5.3	336.3 (*ε*_0.5_)	1
CrMnFeCoNiSn_0.5_ [64]	>100	FCC	L2_1_	1023–1248	10^−3^–10	1683	4.6–9.4	322.2 (*ε*_0.7_)	1
CrMnFeCoNiC_0.5_ [37]	600	FCC	-	973–1273	10^−3^–1	1801	8–26	479 (*ε*_0.6_)	1
CrMnFeCoNi-1 at.%C [20]	125	FCC	M_7_C3	973–1273	10^−3^–1	1801	8.4–33	605.1 (*ε*_0.6_)	1
(CrMnFeCoNi)_95_C_5_ [38]	50	FCC	M_23_C_6_	1073&1273	10^−3^–10^−1^	1801	6.3–10.5	424.1 (*ε*_0.6_)	1
Cr_25_Mn_15_Fe_10_Co_35_Ni_15_ [60]	190	FCC	-	1123–1273	10^−3^–10^−1^	1801	5–6.3	310.4 (*ε*_1.0_)	1
CrFeCoNi [96]	>100	FCC	-	1173–1373	10^−3^–10^−1^	1872	5.8–7	390.1 (*ε*_0.8_)	2
Al_0.3_CrFeCoNi [25]	>100	FCC	-	1023–1223	5 × 10^−4^–10^−1^	1806	5.2–9.7	361.4 (*ε*_0.7_)	2
Al_0.3_CrFeCoNi [92]	52	FCC	-	1023–1423	10^−3^–10	1806	4.5–11.1	320.3 (*ε*_0.7_)	2
Al_0.5_CrFeCoNi [13]	>100	FCC	BCC	1173–1473	10^−3^–1	1767	4.4–5.8	296.8 (*ε*_0.8_)	2
Al_0.5_CrFeCoNi [13]	>100	FCC	BCC	1223–1373	10^−3^–1	1767	4.7–4.8	304 (*ε*_0.8_)	2
Al_0.5_CrFeCoNi [84]	>100	FCC	BCC	1173–1373	1.3 × 10^−3^–10^−1^	1767	4.7–6.3	361.8 (*ε*_0.7_)	2
Al_0.6_CrFeCoNi [53]	>100	FCC	BCC	1173–1473	10^−3^–1	1749	4.5–6.4	380 (*ε*_0.6_)	2
Al_0.6_CrFeCoNi [97]	>100	FCC	BCC	1223–1373	10^−3^–1	1749	3.7–4.5	287.5 (*ε*_0.8_)	2
Al_0.7_CrFeCoNi [72]	>100	FCC	BCC	1173–1373	10^−3^–10^−1^	1732	4.1–5.1	375.5 (*ε*_0.6_)	2
Al_0.7_CrFeCoNi [100]	>100	FCC	BCC	1073–1373	10^−2^–10	1732	4.7–6.5	321.3 (*ε*_0.5_)	2
AlCrFeCoNi [51]	194	BCC1	BCC2	1073–1373	10^−3^–1	1684	3.3–4.2	180.6 (*ε*_0.8_)	2
(CrFeCoNi)_90_Zr_10_ [54]	>100	FCC	Ni_2_Zr + Ni_7_Zr_2_	1073–1323	10^−3^–10	1897	3.9–6.6	327.8 (*ε*_0.6_)	2
CrFeCoNiTa_0.395_ [69]	>100	FCC	Laves	1073–1373	10^−3^–1	1999	4.1–6.9	383.5 (*ε*_0.6_)	2
CrFeCoNiNb_0.25_ [89]	>100	FCC	Laves	1073–1273	10^−2^–10	1923	5–9.2	431.9 (*ε*_0.8_)	2
CrFeCoNiMo_0.2_ [18]		FCC	-	973–1373	10^−3^–1	1921	3.4–20	491.2 (*ε*_0.6_)	2
CrFeCoNiCu (as-cast) [90]	>100	FCC1	FCC2	1073–1173	10^−2^–1	1769	3.4–9.7	374.2 (*ε*_0.4_)	2
CrFeCoNiCu (solid-solutionized) [90]	>100	FCC1	FCC2	1073–1173	10^−2^–1	1769	-	-	2
CrFeCoNiCu_1.2_ [66]	>100	FCC1	FCC2	973–1123	10^−3^–1	1753	9.4–25	394.9 (*ε*_0.3_)	2
Cr_25_Fe_15_Co_45_Ni_15–_0.1C [61]	>100	FCC	-	1123–1273	10^−3^–10^−1^	1828	4.8–7.8	479.6 (*ε*_0.5_)	2
Cr_25_Fe_15_Co_45_Ni_15–_0.05N [61]	>100	FCC	-	1123–1273	10^−3^–10^−1^	1828	4.7–6.2	308.5 (*ε*_0.5_)	2
Cr_10_Mn_40_Fe_40_Co_10–_3.3 at.%C [29]	225	FCC	-	1173–1373	10^−2^–1	1727	5.1–10	466.2 (*ε*_0.6_)	3
MnFeCoNiCu [32]	>100	FCC	-	1123–1323	10^−3^–10	1637	3.2–12	510.2 (*ε*_0.7_)	3
CrMn_2_FeNi_2_Cu [95]		FCC1	FCC2	873–1273	10^−3^–10^−1^	1692	7.2–15.5	363.3 (*ε*_0.7_)	3
TiFeCoNiCu [12]	>100	FCC1	FCC2 + BCC + Ti_2_(Ni,Co)	1073–1273	10^−3^–10^−1^	1721	2.2–4.5	426.4 (*ε*_0.7_)	3
AlCrFeCoNi_2.1_ [22]		FCC	BCC	1073–1373	10^−3^–10	1692	3.3–5.2	295.5 (*ε*_0.6_)	3
AlCrFeCoNi_2.1_ [47]		FCC	BCC	1073–1473	10^−3^–10	1692	3–5.9	332.9 (*ε*_0.6_)	3
AlCrFeNiCu [56]	>100	FCC	BCC	1173–1323	10^−3^–1	1602	3.3–4.3	154.4 (*ε*_0.6_)	3
AlFeCoNiCu [41]	>100	FCC	BCC	1173–1373	10^−1^–10	1520	5.6–7.2	328.3 (*ε*_0.5_)	3
Al_5_Ti_3_Cr_15_Mn_10_(FeNi)_67_ [91]	>100	FCC	BCC	1053–1373	10^−2^–10^−1^	1769	4.1–5.3	375.5 (*ε*_0.6_)	3
Mn_5_Co_25_Fe_25_Ni_25_Ti_20_ [52]		FCC	BCC + Ti_2_Ni + Ti_2_Co	1073–1273	10^−3^–1	1791	2.8–4	305.2 (*ε*_0.6_)	3
TiZrNbMoHf [11]	>100	BCC	-	1073–1473	10^−3^–10^−1^	2444	2.9–6.4	431.4 (*ε*_0.6_)	3
TiZrNbMoHf [40]	>100	BCC	-	1373–1523	10^−3^–5 × 10^−1^	2444	3.2–5.1	290.7 (*ε*_0.6_)	3
Ti_29_Zr_24_Nb_23_Hf_24_ [74]	361	BCC	-	973–1373	10^−3^–10	2308	4.1–10	234.6 (*ε*_0.2_)	3
TiZrNbHfTa [33]	140	BCC	-	1273–1473	10^−4^–10^−2^	2523	2.7–3.3	244.4 (*ε*_0.6_)	3
AlCrFeNi [93]	>100	BCC(A2)	BCC(B2)	1073–1373	10^−3^–1	1663	4.1–8.3	370 (*ε*_0.6_)	3
VNbMoTa [68]	>100	BCC	-	1173–1373	10^−3^–10^−1^	2780	14.5–19.5	454.2 (*ε*_0.5_)	3
AlTiVNb_2_ [79]	>100	BCC(B2)	-	1273–1473	10^−3^–10^−1^	2111	4.1–6.7	391.4 (*ε*_0.6_)	3
AlTi_3_VZr_1.5_Nb [99]	>100	BCC(B2)	-	1373–1523	10^−3^–1	1984	2.9–3	210.6 (*ε*_0.6_)	3
AlCrFeCoNiCu [8]	>100	BCC	FCC	973–1303	10^−3^–10^−1^	1630	2.9–4.4	300.7 (*ε*_0.7_)	3
TiVNbMoTa [71]	0.58	BCC	FCC	1373–1573	5 × 10^−4^–5 × 10^−1^	2612	2.2–3	291.1 (*ε*_0.3_)	3

**Table 2 materials-16-00919-t002:** The constitutive equations of deformation mechanisms in HEAs.

Creep Process	Equation	*k*_i_ Value		
		Ruano et al. [113]	CrMnFeCoNi [62]	Al_0.5_CrMnFeCoNi [62]
Diffusional creep				
Nabarro–Herring [105,106]	${\dot{\varepsilon }}_{1}=k_{1}\left(D_{L}/d^{2}\right)\left(Eb^{3}/kT\right)\left(\sigma /E\right)$	14	14	14
Coble [107]	${\dot{\varepsilon }}_{2}=k_{2}\left(D_{\mathrm{gb}}b/d^{3}\right)\left(Eb^{3}/kT\right)\left(\sigma /E\right)$	50	50	50
Grain boundary sliding (GBS)				
Lattice-diffusion-controlled [109]	${\dot{\varepsilon }}_{3}=k_{3}\left(D_{L}/d^{2}\right){\left(\sigma /E\right)}^{2}$	6.4 × 10^9^	3.1 × 10^8^	6.7 × 10^8^
Pipe-diffusion-controlled [110]	${\dot{\varepsilon }}_{4}=k_{4}\alpha \left(D_{p}/d^{2}\right){\left(\sigma /E\right)}^{4}$	3.2 × 10^11^	1.6 × 10^10^	3.4 × 10^10^
Grain-boundary-diffusion-controlled [109]	${\dot{\varepsilon }}_{5}=k_{5}\left(D_{\mathrm{gb}}b/d^{3}\right){\left(\sigma /E\right)}^{2}$	5.6 × 10^8^	1.9 × 10^7^	5.9 × 10^7^
Slip creep				
Harper–Dorn [108]	${\dot{\varepsilon }}_{6}=k_{6}\left(D_{L}/b^{2}\right)\left(Eb^{3}/kT\right)\left(\sigma /E\right)$	1.7 × 10^−11^	1.7 × 10^−11^	1.7 × 10^−11^
Lattice-diffusion-controlled dislocation climb creep [112]	${\dot{\varepsilon }}_{7}=k_{7}\left(D_{L}/b^{2}\right){\left(\sigma /E\right)}^{5}$	1 × 10^11^	2.6 × 10^9^	1.5 × 10^9^
Pipe-diffusion-controlled dislocation climb creep [112]	${\dot{\varepsilon }}_{8}=k_{8}\left(D_{p}/b^{2}\right){\left(\sigma /E\right)}^{7}$	5 × 10^12^	1.1 × 10^9^	3.9 × 10^9^
Solute drag creep [111]	${\dot{\varepsilon }}_{\mathrm{CJ}}=\frac{2\gamma \tilde{D}kT}{X_{s}X_{a}e^{2}Eb^{5}}{\left[2\left(1+\nu \right)\right]}^{4}\left(\frac{1-\nu }{1+\nu }\right){\left(\frac{\sigma }{E}\right)}^{3}$			

## Data Availability

The raw/processed data required to reproduce these findings cannot be shared at this time, as the data also form part of an ongoing study.

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
