# Peer review of "Deformation Mechanisms and Processing Maps for High Entropy Alloys (Presentation of Processing Maps in Terms of Zener–Hollomon Parameter): Review"

_materials, 2023, doi:10.3390/ma16030919_

Round 1
Reviewer 1 Report
1. The plagiarism percentage is very high. The report is attached. Authors are requested to rewrite.
2. There are many figures from other publishers (such as Elsevier, springer, etc.) But there is no evidence of permissions from the publisher.
3. The authors should include more schematics, flow charts etc. comparing the available literature. This is really required.
4. Use of tables comparing various deformation mechanisms in various alloys to be reported in addition to existing tables.

Author Response
I really appreciate the reviewer’s valuable comments.
- The plagiarism percentage is very high. The report is attached. Authors are requested to rewrite.
--The plagiarism percentage for the revised manuscript is only 5%.
- There are many figures from other publishers (such as Elsevier, springer, etc.) But there is no evidence of permissions from the publisher.
--All the figures are replotted (with variation) or newly created. Thus, no permission from the publishers is required.
- The authors should include more schematics, flow charts etc. comparing the available literature. This is really required.
-- The currently provided data are enough for achieving our goal in this work.
- Use of tables comparing various deformation mechanisms in various alloys to be reported in addition to existing tables.
--The deformation mechanisms of HEAs are not explicitly mentioned in most of the published papers.
Reviewer 2 Report
The manuscript reviews the deformation mechanisms reported in the literature for FCC , BCC and FCC+BCC high entropy alloys in the AlCoCrFeMnNi and AlCoCrFeNi systems. The works includes a large compilation and subsequent analysis of many data reported in the literature for these HEAs, which could be very useful for the readers. The manuscript is well written and the data properly analyzed. Consequently, the manuscript is suitable for publishing although there are some minor points that should be corrected in a new version of the manuscript:
1) The authors state that the atomic size of Al is smaller than that of transition metals, what the opposite is true. This should be corrected and such part rewritten according to the larger atomic size of Al.
2) It should be clearly indicated in the label of Fig. 8 (dotted or continuous lines) which curves correspond to polynomial or exponential fittings.
3) Figure 9 is confusing for the reader because some symbols are not related to any composition given in the labels
4) Deformation maps in Figure 12 are calculated using an average value of Qc of 317 kJ/mol. Since the range of Qc is very wide (from 150 to 600 kJ/mol), it should be precise to consider the Qc of each alloy for constructing the processing maps of Figure 12.
Author Response
I really appreciate the reviewer’s valuable comments.
#2 The manuscript reviews the deformation mechanisms reported in the literature for FCC , BCC and FCC+BCC high entropy alloys in the AlCoCrFeMnNi and AlCoCrFeNi systems. The works includes a large compilation and subsequent analysis of many data reported in the literature for these HEAs, which could be very useful for the readers. The manuscript is well written and the data properly analyzed. Consequently, the manuscript is suitable for publishing although there are some minor points that should be corrected in a new version of the manuscript:
- The authors state that the atomic size of Al is smaller than that of transition metals, what the opposite is true. This should be corrected and such part rewritten according to the larger atomic size of Al.
-- Sorry for this mistake. “smallest” is changed to “largest”. The correction does not affect other sentences.
- It should be clearly indicated in the label of Fig. 8 (dotted or continuous lines) which curves correspond to polynomial or exponential fittings.
-- Figure 8 is revised.
- Figure 9 is confusing for the reader because some symbols are not related to any composition given in the labels
-- Figure 9 is revised.
4) Deformation maps in Figure 12 are calculated using an average value of Qc of 317 kJ/mol. Since the range of Qc is very wide (from 150 to 600 kJ/mol), it should be precise to consider the Qc of each alloy for constructing the processing maps of Figure 12.
-- Figure 12 is newly plotted considering the Qc of each alloy.
Author Response
I really appreciate the reviewer’s valuable comments.
#3 The paper analyzes and explains the mechanism that governs the mode of plastic deformation under high temperature conditions, based on studies published in the literature.
Although the work is laborious and contains many graphical processing of data collected from published works, some figures are not clear enough and cannot justify the claims related to them.
Figures 4, 7, 9 and 12 need a better processing method to increase clarity, otherwise it is not intelligible.
-- Figures 4, 7, 9 and 12 are revised for increasing clarity. Cluster of many data is inevitable because this data cluster indicates the trend.
Page 15. „According to Prasad et al. [101], the total power, P, absorbed in a material is divided into the power dissipation content (G), which represents the power dissipated by plastic deformation giving rise to a temperature increase in the workpiece, and the power dissipation cocontent (J), which represents the power dissipated by a change in its microstructure, such as dynamic recovery and DRX [101].”
-- power dissipation cocontent (J) is changed to power dissipation co-content (J)
Round 2
Reviewer 1 Report
The authors did not address comments 3 and 4. Hence I do not recommend publishing this version.
Author Response
- Deformation mechanisms have not been explicitly mentioned or discussed in many of the references. Thus, this information cannot be summarized in Table. Nevertheless, n1 value can give hint on deformation mechanism, as discussed in the current paper. In the revised version, n1 value range for each alloy is added in Table 1.